# Characteristics and Dental Indices of Orthodontic Patients Using Aligners or Brackets

**DOI:** 10.3390/ijerph19116569

**Published:** 2022-05-27

**Authors:** Tzu-Han Liao, Jason Chen-Chieh Fang, I-Kuan Wang, Chiung-Shing Huang, Hui-Ling Chen, Tzung-Hai Yen

**Affiliations:** 1Department of Dentistry and Craniofacial Orthodontics, Chang Gung Memorial Hospital, Linkou Branch, Taoyuan 333, Taiwan; angel0219@cgmh.org.tw; 2School of Medicine, College of Medicine, Chung Shan Medical University, Taichung 402, Taiwan; s0601014@gm.csmu.edu.tw; 3Department of Nephrology, China Medical University Hospital, Taichung 404, Taiwan; ikwang@mail.cmuh.org.tw; 4College of Medicine, China Medical University, Taichung 406, Taiwan; 5Craniofacial Research Center, Department Craniofacial Orthodontics, Chang Gung Memorial Hospital, Taipei 105, Taiwan; cshuang@cgmh.org.tw; 6College of Medicine, Chang Gung University, Taoyuan 333, Taiwan; 7Clinical Poison Center, Department of Nephrology, Chang Gung Memorial Hospital, Linkou Branch, Taoyuan 333, Taiwan

**Keywords:** aligners, metal brackets, skeletal relationships, molar relationships, decayed, missing, and filled tooth index, index of complexity outcome and need, Frankfort-mandibular plane angle, E-line of lower lip

## Abstract

Background. Clear aligners have become a treatment alternative to metal brackets in recent years due to the advantages of aesthetics, comfort, and oral health improvement. Nevertheless, few studies have analyzed the clinical characteristics and dental indices of orthodontic patients using aligners or brackets. Methods. A total of 170 patients received orthodontic treatment at Chang Gung Memorial Hospital in 2021. Patients were stratified by types of treatment (Invisalign^®^ clear aligner (*n* = 60) or metal bracket (*n* = 110). Results: Patients were aged 26.1 ± 7.2 years, and most were female (75.0%). The Invisalign^®^ group was older than the bracket group (*p* = 0.003). The skeletal relationships were mainly Class I (49.4%), followed by Class II (30.0%) and Class III (20.6%). The molar relationships were primarily Class I (38.8%), followed by Class II (37.1%) and Class III (24.1%). The decayed, missing, and filled tooth (DMFT) index was 9.9 ± 6.0, including 2.1 ± 2.9 for decayed teeth, 0.5 ± 1.1 for missing teeth, and 7.3 ± 4.3 for filled teeth. There were no significant differences in the DMFT index or skeletal and molar relationships between the groups (*p* > 0.05). The index of complexity outcome and need (ICON) was 56.8 ± 13.5, and the score was lower in the Invisalign^®^ group than in the bracket group (*p* = 0.002). Among the variables included in the ICON assessment, only the aesthetic variable was lower in the Invisalign^®^ group than in the bracket group (*p* < 0.001). The Frankfort-mandibular plane angle was 27.9 ± 5.1 degrees. Finally, the E-line of the lower lip was lower in the Invisalign^®^ group than in the bracket group (1.5 ± 2.4 versus 2.8 ± 3.1, *p* = 0.005). Conclusions. Older patients showed a greater intention to choose Invisalign^®^ treatment for improving the appearance of their teeth than younger patients, who chose metal bracket treatment. The demand for Invisalign^®^ aligner treatment for aesthetic reasons was substantial. A soft tissue profile with more protrusive lower lips and a greater need for orthodontic treatment was found in the bracket group.

## 1. Introduction

There have been increasing numbers of patients seeking clear aligner orthodontic treatments instead of conventional metal brackets in recent years. The development of this computer-aided clear aligner technique has provided another treatment choice to patients for the improvement of aesthetics, comfort, and oral health. A series of removable polyurethane aligners have become a treatment alternative to fixed labial braces due to their biological, aesthetic, and psychological advantages [1]. In 1997, Align Technology, CA, USA was founded, and Invisalign^®^ clear aligner was marketed in 1999 [2,3]. The company fabricates a series of custom-made aligners that sequentially reposition the teeth through computer-aided design and manufacturing concepts. After continually improving the technology, Invisalign^®^ has become one of the most commonly used systems among aligners.

Several studies have focused on assessing the treatment outcome of Invisalign^®^ in comparison to the metal bracket system. Regardless of the limited treatment effect reported on Invisalign^®^-treated patients, there was a statistically similar satisfaction outcome in comparison to bracket-based patients in the assessment of the survey of Pacheco-Pereira et al. [4] in 2018. The Invisalign^®^ group even reported more satisfaction with eating and chewing. Patients choose clear aligners because they have more cosmetic demand. Also, the aligners are associated with less discomfort during treatment, and are more beneficial for oral health [5]. Modern patients choose treatment modalities that focus not only on the treatment effects and treatment expenses, but also on their quality of life during treatment. There are several investigations to assess the quality of different treatment methods in orthodontics [4,6]; few are related to the need before treatment and satisfaction after treatment. Regarding the satisfaction survey, the assessment tools are still limited to questionnaires [7] and are subject to subjectivity. Furthermore, there is still a lack of previous studies focusing on the characteristics of patients who tend to choose clear aligner treatment.

Therefore, the objective of this study is to analyze the clinical characteristics and dental indices of patients who choose clear aligner or metal bracket treatment.

## 2. Materials and Methods

### 2.1. Patients

A total of 170 patients received orthodontic treatment at Chang Gung Memorial Hospital in 2021. After pretreatment data collection, each patient’s data was analyzed, and the treatment plan was determined after discussion with the patient. Then, each patient was thoroughly informed of the advantages and disadvantages of available orthodontic appliances: for example, oral hygiene maintenance, periodontal health, interference of eating, disturbance of speech. The patients were completely independent in choosing either metal brackets (Roth’s prescription, TOMY, Japan, “Genius” self-ligating brackets, MEM, Taiwan and Clippy-C, TOMY, Japan) or clear aligner (Invisalign^®^, Align technology). Demographic records, such as age, sex, height, weight, and BMI, were obtained. Pretreatment intraoral photographs, dental casts, and radiographs were collected to assess the decayed, missing, and filled tooth (DMFT) index, index of complexity outcome and need (ICON), dental malocclusion type, sagittal skeletal relationship, and cephalometric facial dimensions.

### 2.2. Inclusion and Exclusion Criteria

All adult patients who visited the orthodontic clinic of Chang Gung Memorial Hospital in 2021 were recruited for this study. Patients who refused to sign the informed consent form were omitted from the analysis. Furthermore, this study also excluded patients who had previous orthognathic surgery, dentofacial trauma, or craniofacial malformation.

### 2.3. Assessment of the Sagittal Skeletal Relationship

The anterior-posterior skeletal relationship between the maxilla and the mandible of the orthodontic patient was classified as Class I, Class II, or Class III (Figure 1 and Figure 2) according to Steiner’s [8] and Tweed’s [9] analyses (Table 1).

### 2.4. Assessment of Dental Malocclusion Type

The dental malocclusion type of the orthodontic patient was defined as molar Class I, Class II, or Class III based on Angle’s classification (Figure 3).

### 2.5. Assessment of Facial Pattern

The Frankfort–mandibular plane angle (FMA) and sella-nasion to mandibular plane (SN–MP) angle were applied to define three facial patterns in orthodontics (Figure 4 and Figure 5). The average angle of FMA was between 27 degrees and 34 degrees, and FMA smaller than 27 degrees was defined as low angle patient, and FMA larger than 34 degrees was defined as high angle patient.

In addition, Merrifield’s profile line and Z angle [10] (an angle formed by a chin-protrusive line intersecting the Frankfort horizontal plane), Burstone’s B line [11,12] (a line from the chin to the subnasale), Steiner’s S1 line [13,14] (a line from the chin to the midpoint bisecting the nasal nostril border line), Sushner’s S2 line [15] (a line from the soft tissue nasion to soft tissue pogonion), Holdaway’s H line [16,17] (a line from the chin to the upper lip), and Ricketts’ E-line [18,19] were all suitable to analyze facial configurations. Ricketts’ E-lines [18,19] and Z angle [10] were measured in this study. The Ricketts’ E-line is one of most convenient reference lines, in comparison to other reference lines, because of its anterior location [20]. Additionally, the Z angle provides a critical description of the lower face relationship and eliminates the vagueness of “eye judgment”; it is the angle between the Frankfort–mandibular plane angle and soft tissue profile (Figure 6), which quantifies facial balance. Lip protrusion is defined by the relationship of the upper and lower lips to the Ricketts’ E-line. The Ricketts’ E-line is a line drawn from the tip of the nose to the soft-tissue pogonion (Figure 7) [18]. The upper lip and lower lip should be tangent to this line for Caucasian individuals. On the other hand, the upper and lower lip are more protrusive in Chinese individuals, and a more convex facial profile is seen compared with Caucasian individuals [21]. The Z angle is another angular measurement for evaluating the soft tissue profile of the esthetics of the lower face. It is formed by the intersection of the Frankfort horizontal plane and the profile line, which is established by drawing a line from the soft-tissue chin to the most anterior point of either the lower or the upper lip, whichever protrudes the most. The normal range of the Z angle is typically 70–80 degrees [10].

### 2.6. Assessment of the DMFT Index

The DMFT index is the sum of the numbers of decayed teeth, teeth missing due to caries, and filled teeth in the permanent teeth. It is an indicator used to measure an individual’s dental caries situation, and it further reflects the deterioration of oral hygiene.

### 2.7. Assessment of the ICON

The ICON is an index of treatment need and an occlusal index of malocclusion complexity. It was proposed by Daniels C and Richmond S in 2000 [22] and it could evaluate not only the treatment need but also the treatment outcome (Figure 8). The ICON is composed of assessments of dental aesthetics, upper arch crowding, the presence of crossbite, the anterior vertical relationship (incisor overbite), and the buccal segment anteroposterior interdigitation. The aesthetic component involves comparing the frontal photo of the dentition to the illustrated scale [22], and the scale is graded from 1 (most attractive) to 10 (least attractive). By analyzing the study model, the other occlusal traits are scored as numeric values according to the standard protocol [22]. Each item is attached to a different weight, and the final ICON score is divided into malocclusion complexity grades (<29 = easy, 29–50 = mild; 51–63 = moderate, 64–77 = difficult, >77 = very difficult) [22]. A cutoff point of 43 is set to determine a definite need for orthodontic treatment.

### 2.8. Statistical Analysis

For descriptive statistics, all continuous variables were presented as the means and standard deviations, and categorical variables were presented as numbers and percentages. Comparisons between the Invisalign and bracket groups were performed using independent t tests for quantitative variables and chi-square tests for categorical variables. All statistics were calculated using IBM SPSS Statistics for Windows, Version 25.0. (Armonk, NY, USA, 2017). The significance level for all tests was set at *p* value < 0.05.

## 3. Results

### 3.1. Baseline Demographics

A total of 170 patients received orthodontic treatment at Chang Gung Memorial Hospital in 2021 (Table 2). Patients were stratified by types of orthodontic device (Invisalign^®^ clear aligner (*n* = 60) or metal brackets (*n* = 110)). The patients were aged 26.1 ± 7.2 years, and most were female (75.0%). The Invisalign^®^ group was older than the bracket group (28.6 ± 8.5 versus 24.8 ± 6.1, *p* = 0.003). No significant differences in other demographic variables were noted between the groups (*p* > 0.05).

### 3.2. Sagittal Skeletal Relationship

Table 3 showed that the skeletal relationships were mainly Class I (49.4%), followed by Class II (30.0%) and Class III (20.6%). No significant difference was noted between the groups (*p* = 0.558).

### 3.3. Dental Malocclusion Type

A total of 38.8% of the study subjects presented a Class I molar relationship, 37.1% presented a Class II molar relationship, and 24.1% presented a Class III molar relationship (Table 4). No significant difference was noted between the groups (*p* = 0.912).

### 3.4. Facial Pattern

The FMA was 27.9 ± 5.1 degrees, which was within the average range of Taiwanese people (Table 5). The Invisalign^®^ group presented a less protrusive lower lip than the brackets group (Ricketts’ E-line to the lower lip, Invisalign^®^: 1.5 ± 2.4 mm; bracket: 2.8 ± 3.1 mm, *p* = 0.005). No significant differences were noted between the groups for other facial variables.

### 3.5. Decayed, Missing, and Filled Tooth (DMFT) Index

The DMFT index was 9.9 ± 6.0, including 2.1 ± 2.9 for decayed teeth, 0.5 ± 1.1 for missing teeth, and 7.3 ± 4.3 for filled teeth (Table 6). No significant differences were noted between the groups.

### 3.6. Index of Complexity Outcome and Need (ICON)

The index of complexity outcome and need (ICON) was 56.8 ± 13.5, and the score was lower in the Invisalign^®^ group than in the bracket group (*p* = 0.002, Table 7). Among the variables included in the ICON assessment, only the aesthetic variable was significantly lower in the Invisalign^®^ group than in the bracket group (*p* < 0.001).

## 4. Discussion

Most of the orthodontic patients were young (26.1 ± 7.2 years), and most of them were female (75.0%). Older patients showed more intention to choose Invisalign^®^ treatment for improving the appearance of their teeth than younger patients, who chose metal bracket treatment. Young single females (aged 18–30 years) with higher incomes were significantly more likely to seek orthodontic treatment according to the Medical Expenditure Panel Survey in the United States, 2000–2004 [23]. The mean age of the Taiwanese patients seeking orthodontic treatment was approximately 25.2 years [24], which was similar to the average age (24.8 ± 6.1) in the bracket group. Nevertheless, the Invisalign^®^ group (28.6 ± 8.5 years) was slightly older than the bracket group (24.8 ± 6.1 years). The possible explanation might be that there were more students recruited into the bracket group, and most were not financially independent. The higher price of the Invisalign^®^ treatment might pose a potential burden to patients who have a limited budget.

There were no significant differences between the Invisalign^®^ and bracket groups in terms of dental and skeletal variables. In addition, there was no significant difference in the DMFT index between the groups. The mean DMFT index of our patients was 9.9 ± 6.0. In comparison to the DMFT index in other countries, for example, 6.2 in the United States, 7.3 in Iran, 12.5 in Spain, 12.28 in Japan, and 12.10 in Malaysia [25,26,27], our data was within the ranges. Radiographs are more sensitive in detecting incipient caries and restorations than visual inspection [28]. In our study, we used bitewings and periapical radiographs to detect decayed and filled teeth, which might be one of the reasons why our patients exhibited a slightly higher DMFT index than those in other studies.

The mean ICON score of all patients was 56.8 ± 13.5. The ICON score was 58.9 in South Asia, 57.3 in the United States, 56.6 in Australia, 55.8 in Africa, 52.4 in the Middle East, 50.3 in Europe, 47.2 in North and East Asia [29], and 44.6 in Iran [30]. Our data was similar to these scores. There was a significant difference in ICON score between the Invisalign^®^ and bracket groups (59.4 ± 12.5 versus 65.6 ± 12.6, *p* = 0.002). According to Daniels et al. [22], the Invisalign^®^ group was classified as moderate complexity (51–63), whereas the bracket group was classified as difficult complexity (64–77). The components that influenced the different ICON scores of the two groups mostly arose from the aesthetic component (6.0 ± 1.3 versus 6.8 ± 1.2, *p* < 0.001). The aesthetic component was heavily weighted. The aesthetic assessment involved comparing the dentition to the illustrated scale, considering similar outcomes, including overjet, anterior crossbite, upper and lower incisor inclination, lip incompetence, and gingival display. Increasing numbers of patients are asking for clear aligner orthodontic treatments instead of conventional metal brackets because of their aesthetic appearance and reduced discomfort, and their benefits for oral hygiene maintenance. Based on a survey conducted by Meier et al. [2] about the characteristics of patients who were interested in Invisalign^®^, 97% reported aesthetic considerations as their primary reason. Accordingly, we might assume that patients seeking Invisalign^®^ treatment were more concerned about aesthetics. This might be related to our findings that patients in the Invisalign^®^ group had significantly less severe aesthetic problems than those in the bracket group. In other words, the Invisalign^®^ group had a greater perception of attractiveness, especially in terms of a better smile, including dental and soft tissue harmony.

The aesthetic score in the ICON is more opinion-based and is influenced by bias in personal preference. The measurement of the Z angle and the distance of the lips to the Ricketts’ E-line provide more objective data to justify a pleasing facial profile. The lower lip to Ricketts’ E-line was 2.3 ± 2.9 mm on average for all patients in our study, in comparison to 1.7 mm in Chinese patients, −0.3 mm in Japanese patients, 1 mm in Korean patients, and −5.0 mm in Caucasian patients [21]. Taiwanese individuals were found to have a shorter nose but a thicker upper lip and chin than Caucasian individuals [31] through research involving different ethnic groups. Our patients showed a protrusive lower lip, which might be one of the reasons why they sought orthodontic treatment. Lip prominence is thought to be an unattractive trait and an unsatisfactory situation, particularly in adults [18]. In addition, the significantly more protrusive lower lip relating to the Ricketts’ aesthetic line in the bracket group, than in the Invisalign^®^ group, might coincide with the significant difference in aesthetic scores of the ICON between these two groups. The Ricketts’ E line was a reference line from the chin to the tip of the nose [18]. Thus, the difference between the two groups might be due to variations in the nose, lip, and chin. Protrusive incisors, a large overjet, inadequate lip length, lip thickness, the position of the chin, and the height of the nose, or a combination, might influence the relationship of the lower lip to the Ricketts’ E line [32]. Further investigations and measurements might be needed to analyze possible variables for differentiation between the two groups of patients.

The analytical data is valuable and is of importance to clinical practice. First, we found in this study that patients who were high-aesthetic demanding might have increased intentions to obtain the clear aligner treatment. Instead of focusing on the occlusion, function, etc., the patients seeking the clear aligner treatment tended to take aesthetics as a primary treatment goal. Understanding a patient’s treatment goal is the cornerstone to achieving a satisfied treatment outcome. Second, in discussing more details regarding the components of aesthetics, as we mentioned earlier, this mainly represented the balance of the lower anterior face, including nose, mouth, teeth, lips, and chin. Smile improvement over the anterior area is an important motivator for those who seek orthodontic treatment [33]. However, aesthetics was very subjective, and it varied among different ages, genders, and races. The simulation of treatment outcome with the aid of the 3D computer-aided design clear aligner technique helps to ensure the clinician and the patient both have the same perception of treatment outcome. In addition, the technology provides precisely customized modification, which helps to match a patient’s preference. Finally, the patients with increased intentions and performing behaviors were found to be significantly associated with visiting the orthodontist regularly and being more cooperative during orthodontic treatment [34]. Patient adherence and cooperation played decisive roles in the orthodontic treatment, when estimating treatment duration. In summary, this study compared the patient characteristics between the clear aligner and the bracket group. After realizing aesthetics was the main treatment objective of the patient, the clinician suggested focusing more on the balance of lower anterior face, especially anterior teeth. Proper achievement of the patient’s main concern might help to improve patient compliance and confidence, which were critical in the clear aligner treatment. Eventually, a win-win treatment result would be obtained.

The main limitation of this study was the small sample size, and the samples were only recruited from Chang Gung Memorial Hospital, which might be insufficient to represent the general population in Taiwan. For example, nearly 35% of patients decided to have an aligner instead of brackets as an orthodontic appliance in our study, which was remarkably higher than <20% in North America and <10% worldwide [35,36]. Therefore, further studies with larger, more representative, samples are suggested to understand the clinical characteristics of orthodontic patients choosing clear aligners or metal brackets. Besides this, the overall well-being and oral hygiene habits, such as daily oral hygiene practice, could also influence patients’ choices of clear aligner or conventional metal bracket. Therefore, lacking the quality-of-life questionnaires and periodontal tissues examination are also limitations of our study.

## 5. Conclusions

Most of the orthodontic patients in this study were young females. Older patients showed more intention to choose Invisalign^®^ treatment for improving the appearance of their teeth than younger patients, who chose metal bracket treatment. The dental and skeletal malocclusions and facial patterns were similar in both groups. The demand for Invisalign^®^ aligner treatment for aesthetic reasons was substantial. A soft tissue profile with a more protrusive lower lip and more need for orthodontic treatment were found in the bracket group.

## Figures and Tables

**Figure 1 ijerph-19-06569-f001:**
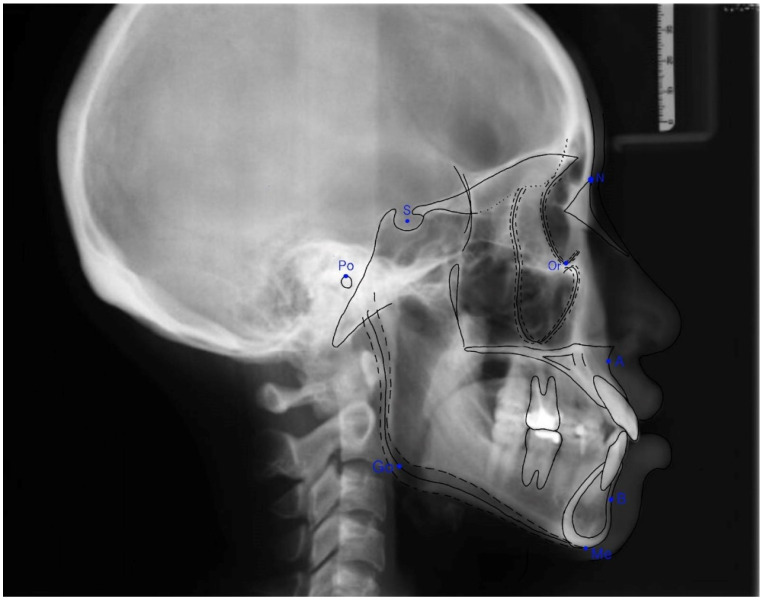
The landmarks used to analyze lateral cephalometric radiographs.

**Figure 2 ijerph-19-06569-f002:**
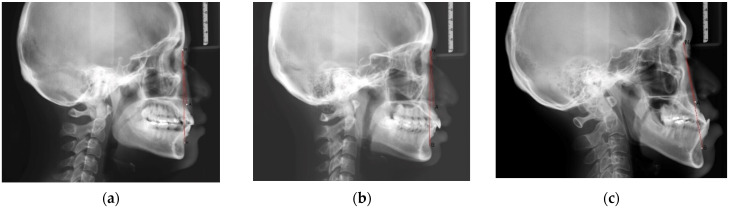
The sagittal skeletal relationship was classified as Class I, Class II, or Class III based on the ANB angle, which was formed by the A point, Nasion, and B point: (**a**) Class I relationship: ANB angle was 2–4 degrees; (**b**) Class II relationship: ANB angle was larger than 4 degrees; (**c**) Class III relationship: ANB angle was smaller than 2 degrees.

**Figure 3 ijerph-19-06569-f003:**
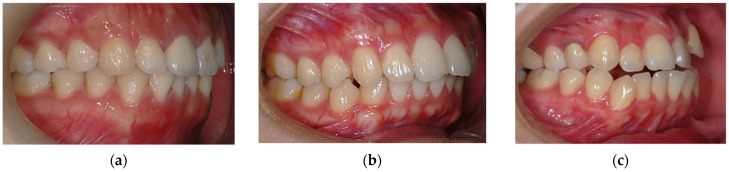
Dental malocclusion based on Angle’s classification: (**a**) Class I molar relationship: the mesiobuccal cusp of the upper first molar occludes the mesiobuccal groove of the lower first molar; (**b**) Class II molar relationship: the mesiobuccal cusp of the upper first molar occludes in front of the mesiobuccal groove of the lower first molar; (**c**) Class III molar relationship: the mesiobuccal cusp of the upper first molar occludes behind the mesiobuccal groove of the lower first molar.

**Figure 4 ijerph-19-06569-f004:**
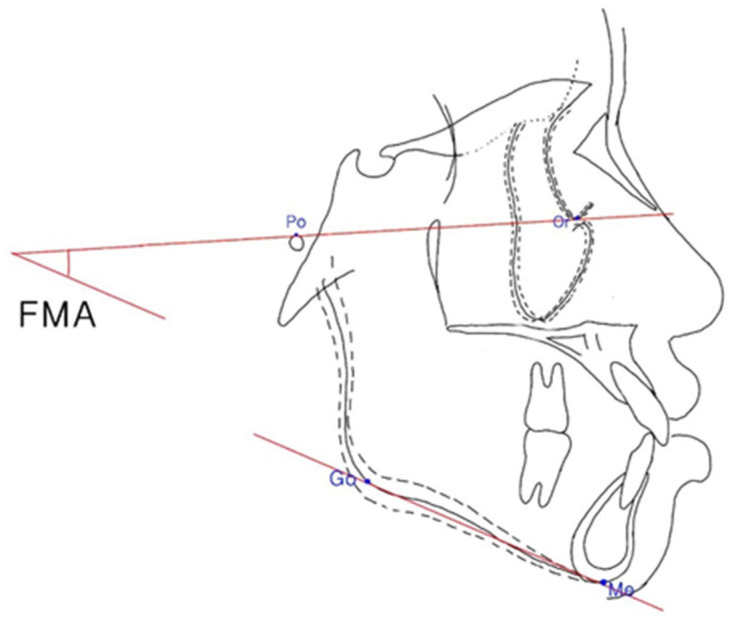
Frankfort–mandibular plane angle (FMA). The FMA was constructed by the intersection of the Frankfort horizontal plane and the mandibular plane.

**Figure 5 ijerph-19-06569-f005:**
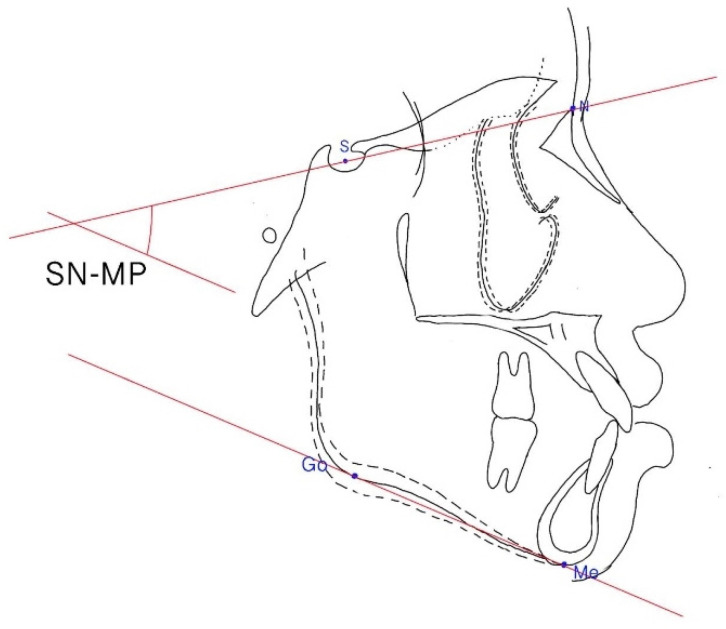
Sella-nasion plane to the mandibular plane (SN–MP) angle. The SN-MP angle was formed by the intersection of the SN plane and the mandibular plane.

**Figure 6 ijerph-19-06569-f006:**
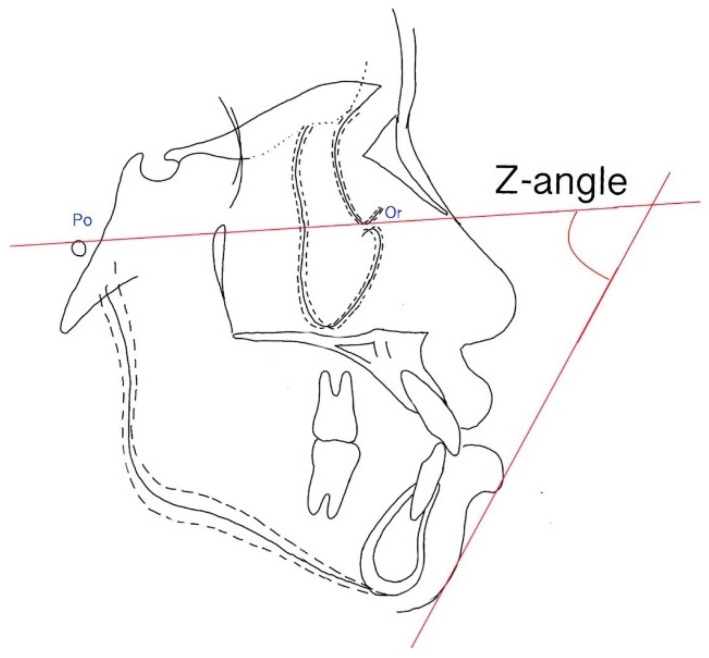
Z angle. The angle is formed by the Frankfort horizontal plane angle and soft tissue profile.

**Figure 7 ijerph-19-06569-f007:**
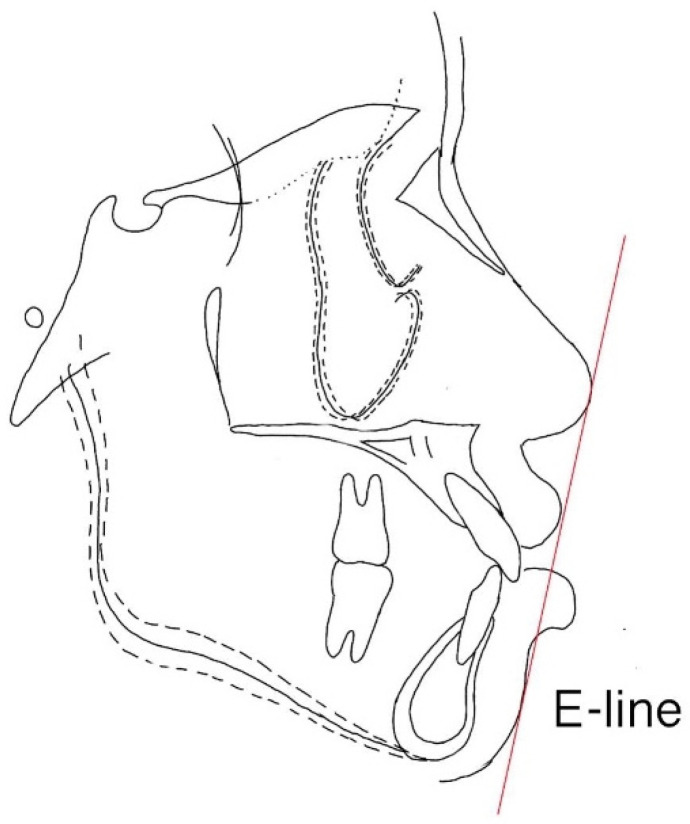
Ricketts’ Esthetic line (E-line). The line is drawn from the tip of the nose to the soft-tissue pogonion.

**Figure 8 ijerph-19-06569-f008:**
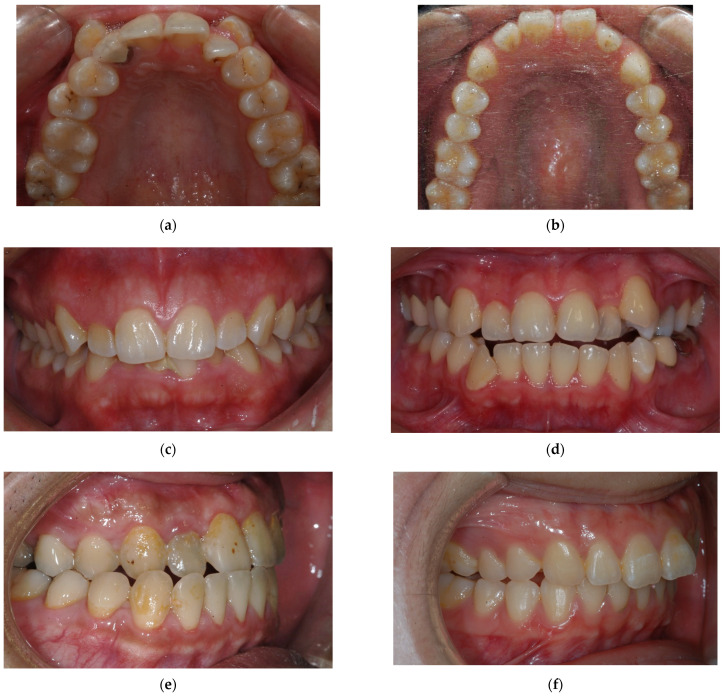
(**a**,**b**) Upper arch crowding/spacing: (**a**) Crowding indicates that there is a discrepancy between the space required by the teeth and the space available. (**b**) Spacing: the opposite of crowding, the space available exceeds the space needed. (**c**,**d**) Incisor overbite: Incisor overbite indicates the anterior vertical relationship. (**c**) Deep bite: there is a vertical overlap of the incisors. (**d**) Open bite: there is no vertical overlap of the incisors. (**e**) Crossbite: Crossbite indicates the transverse relationship. Normally, the buccal cusp of the mandibular dentition should occlude to the lingual cusp of the maxillary dentition. (**f**) Buccal segment anteroposterior relationship: The anteroposterior cuspal relationship of premolars and molars. Normally, it should be a cusp-to-embrasure relationship.

**Table 1 ijerph-19-06569-t001:** Definitions of terms used in assessing the sagittal skeletal relationship.

Landmark	Definition
Sella (S)	The center of the hypophyseal fossa (sella turcica)
Nasion (N)	The junction of the nasal and frontal bones at the most posterior point on the curvature of the bridge of the nose
Porion (Po)	The uppermost point of the external ear meatus
Orbitale (Or)	A point midway between the lowest point on the inferior margin of the two orbits
Menton (Me)	The lowest point on the symphysis of the mandible
A point (A)	The innermost curvature of the maxillary apical base
B point (B)	The innermost curvature from the chin to the alveolar junction

**Table 2 ijerph-19-06569-t002:** Baseline demographics of patients, stratified by orthodontic device (Invisalign^®^ or bracket) (*n* = 170).

Variable	All Patients (*n* = 170)	Invisalign^®^ Group (*n* = 60)	Bracket Group (*n* = 110)	*p* Value
Age (year)	26.1 ± 7.2	28.6 ± 8.5	24.8 ± 6.1	0.003 **
Female, *n* (%)	127 (75%)	41 (68.3%)	86 (78.2%)	0.176
Height (cm)	162.9 ± 7.2	164.0 ± 7.1	162.2 ± 7.3	0.129
Weight (kg)	56.6 ± 10.6	58.0 ± 11.0	55.9 ± 10.3	0.227
Body mass index (kg/m^2^)	21.2 ± 3.2	21.4 ± 3.0	21.1 ± 3.3	0.631

Note: ** *p* < 0.01.

**Table 3 ijerph-19-06569-t003:** Sagittal skeletal relationship of patients, stratified by orthodontic device (Invisalign^®^ or bracket) (*n* = 170).

Variable	All Patients (*n* = 170)	Invisalign^®^ Group (*n* = 60)	Bracket Group (*n* = 110)	*p* Value
Class I, *n* (%)	84 (49.4)	32 (38.1)	52 (61.9)	0.558
Class II, *n* (%)	51 (30.0)	17 (33.3)	34 (66.7)	
Class III, *n* (%)	35 (20.6)	11 (31.4)	24 (68.6)	

**Table 4 ijerph-19-06569-t004:** Malocclusion type of patients, stratified by orthodontic device (Invisalign^®^ or bracket) (*n* = 170).

Variable	All Patients (*n* = 170)	Invisalign^®^ Group (*n* = 60)	Bracket Group (*n* = 110)	*p* Value
Class I, *n* (%)	66 (38.8)	23 (34.8)	43 (65.2)	0.912
Class II, *n* (%)	63 (37.1)	22 (34.9)	41 (65.1)	
Class III, *n* (%)	41 (24.1)	15 (36.6)	26 (63.4)	

**Table 5 ijerph-19-06569-t005:** Cephalometric facial measurements of patients, stratified by orthodontic device (Invisalign^®^ or bracket) (*n* = 170).

Variable	All Patients (*n* = 170)	Invisalign^®^ Group (*n* = 60)	Bracket Group (*n* = 110)	*p* Value
FMA	27.9 ± 5.1	27.3 ± 4.8	28.3 ± 5.3	0.218
SN-MP	36.6 ± 7.7	36.6 ± 10.3	36.5 ± 5.8	0.960
Z angle	68.1 ± 12.7	66.6 ± 12.0	68.9 ± 13.1	0.245
E-line, upper lip	0.1 ± 2.8	0.1 ± 2.5	0.1 ± 3.0	0.951
E-line, lower lip	2.3 ± 2.9	1.5 ± 2.4	2.8 ± 3.1	0.005 **

Note: ** *p* < 0.01, FMA, Frankfort–mandibular plane angle, SN-MP, SN plane to the mandibular plane.

**Table 6 ijerph-19-06569-t006:** Decayed, missing, and filled tooth (DMFT) index of patients, stratified by orthodontic device (Invisalign^®^ or bracket) (*n* = 170).

Variable	All Patients (*n* = 170)	Invisalign^®^ Group (*n* = 60)	Bracket Group (*n* = 110)	*p* Value
Decayed	2.1 ± 2.9	2.3 ± 3.1	1.9 ± 2.6	0.391
Missing	0.5 ± 1.1	0.5 ± 0.9	0.5 ± 1.2	0.696
Filled	7.3 ± 4.3	7.1 ± 4.7	7.4 ± 4.1	0.682
DMFT index	9.9 ± 6.0	9.9 ± 6.0	9.9 ± 6.1	0.969

**Table 7 ijerph-19-06569-t007:** Index of complexity outcome and need (ICON) of patients, stratified by orthodontic device (Invisalign^®^ or bracket) (*n* = 170).

	All Subjects (*n* = 170)	Invisalign (*n* = 60)	Brackets (*n* = 110)	*p* Value
Aesthetic	6.5 ± 1.2	6.0 ± 1.3	6.8 ± 1.2	<0.001 ***
Upper arch crowding	1.1 ± 1.2	1.0 ± 1.1	1.2 ± 1.2	0.246
Crossbite	0.4 ± 0.5	0.4 ± 0.5	0.5 ± 0.5	0.570
Incisor overbite	0.8 ± 0.8	0.7 ± 0.9	0.8 ± 0.8	0.422
Buccal segment anteroposterior	2.2 ± 0.8	2.0 ± 0.7	2.2 ± 0.8	0.425
ICON	56.8 ± 13.5	59.4 ± 12.5	65.6 ± 12.6	0.002 **

Note: ** *p* < 0.01, *** *p* < 0.001.

## Data Availability

The datasets used and analyzed for this study are available from the corresponding author upon request.

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
