# Peer review of "Characteristics and Dental Indices of Orthodontic Patients Using Aligners or Brackets"

_ijerph, 2022, doi:10.3390/ijerph19116569_

Round 1
Reviewer 1 Report
I also think that the topic may be of interest to the field.
I have a few relatively minor comments that the authors might consider.
①Please describe the purpose of the study definitely.
②Please specify that you got the approval of the ethic screening committee.
Reviewer 2 Report
The study investigated the characteristics of patients who choose clear aligners or conventional metal bracket treatments. By comparing the demographic profiles, DMFT index, ICON, dental malocclusion type, sagittal skeletal relationship, and cephalometric facial dimensions in these two groups, the authors concluded that age, aesthetic reasons, and soft tissue profile can affect patients’ preferences.
The study is interesting, the objective of the study is clear, and many aspects were evaluated.
However, I have a concern that there may be other important factors that could influence the patients’ choices, such as overall well-being and oral hygiene habits. It would be more comprehensive if the study includes some questionnaires such as Quality-of-life questionnaires, or periodontal tissues examination.
Also, there are several typo errors.
Author Response
The study investigated the characteristics of patients who choose clear aligners or conventional metal bracket treatments. By comparing the demographic profiles, DMFT index, ICON, dental malocclusion type, sagittal skeletal relationship, and cephalometric facial dimensions in these two groups, the authors concluded that age, aesthetic reasons, and soft tissue profile can affect patients’ preferences.
The study is interesting, the objective of the study is clear, and many aspects were evaluated.
Response: Thank you for reviewing our manuscript.
However, I have a concern that there may be other important factors that could influence the patients’ choices, such as overall well-being and oral hygiene habits. It would be more comprehensive if the study includes some questionnaires such as Quality-of-life questionnaires, or periodontal tissues examination.
Response: Thank you for the comments. We understand that the overall well-being and oral hygiene habits could influence patients’ choices of clear aligner or conventional metal bracket. We apologize for the study design (as approved by Institutional Review Board, coded 201900197B0) is incomplete because quality-of-life questionnaires and periodontal tissues examination are not included. These important limitations have been stressed in the Discussion section.
Also, there are several typo errors.
Response: Thank you for reminding us. The manuscript has been submitted to American Journal Experts for language editing before submission to IJERPH. We have reviewed the manuscript once again and spotted the typo errors. Please advise us again if we still miss out on anything.

Reviewer 3 Report
This article focuses on the description of multiple parameters (socio-demographic, radiographic, clinical ...) to map the differences between individuals (Invisalign vs brackets).
The article is well presented, well written and pleasant to read. The study has been well conducted.
I have no major concern except one: I really have trouble seeing the point of this study. It is a description of a population in a clinical practice. That part is very well done. But what is the concrete message for the clinician? Why this study could change her/his practice or draw her/his attention to a particular point? I think that this practical part is sorely lacking in the discussion. What are the important messages to remember?
Author Response
This article focuses on the description of multiple parameters (socio-demographic, radiographic, clinical ...) to map the differences between individuals (Invisalign vs brackets).
Response: Thank you for reviewing our article.
The article is well presented, well written and pleasant to read. The study has been well conducted.
Response: Thank you for the comments.
I have no major concern except one: I really have trouble seeing the point of this study. It is a description of a population in a clinical practice. That part is very well done. But what is the concrete message for the clinician? Why this study could change her/his practice or draw her/his attention to a particular point? I think that this practical part is sorely lacking in the discussion. What are the important messages to remember?
Response: Thank you for the suggestions. We apologized for inadequate focus on the information regarding clinical application and influence on daily practice, so we emphasized more on the revised Discussion section.
The analytical data is valuable and is of importance to clinical practice. First, we found in this study that patients who were high-aesthetic demanding might have increased intentions to obtain the clear aligner treatment. Instead of focusing on the occlusion, function, etc, the patients seeking for the clear aligner treatment tended to take aesthetics as a primary treatment goal. Understanding a patient's treatment goal is the cornerstone to achieve a satisfied treatment outcome. Second, to discuss more details regarding the components of esthetics, as we mentioned earlier, it mainly represented the balance of the lower anterior face, including nose, mouth, teeth, lip, and chin. Smile improvement over the anterior area is an important motivator for those who seek orthodontic treatment. However, aesthetics was very subjective, and it varied from different ages, genders, and races. The simulation of treatment outcome by the aid of the 3D computer-aided design clear aligner technique helps to ensure the clinician and the patient both have the same perception of treatment outcome. In addition, the technology provides precisely customized modification, which helps to match a patient's preference. At last, the patients with increased intentions and performing behaviors were found significantly associated with visiting the orthodontist regularly and more cooperative during orthodontic treatment. The patient adherence and the cooperation were decisive in the orthodontic treatment, while estimating the treatment duration. In summary, this study compared the patient characteristics between the clear aligner and the bracket group. After realizing the aesthetic was the main treatment objective of the patient, the clinician was suggested to focus more on the balance of lower anterior face, especially anterior teeth. Proper achievement of the patient's main concern might help to improve the patient compliance and confidence, which was critical in the clear aligner treatment. Eventually, a win-win treatment result would be obtained.

Round 2
Reviewer 3 Report
The modifications proposed by the authors substantially improve the interest of the results of the article